# The Effects of Antimicrobial Protocols and Other Perioperative Factors on Postoperative Complications in Horses Undergoing Celiotomy: A Retrospective Analysis, 2008–2021

**DOI:** 10.3390/ani13223573

**Published:** 2023-11-19

**Authors:** Meagan Rockow, Gregg Griffenhagen, Gabriele Landolt, Dean Hendrickson, Lynn Pezzanite

**Affiliations:** Department of Clinical Sciences, College of Veterinary Medicine and Biomedical Sciences, Colorado State University, Fort Collins, CO 80523, USA; meagan.rockow@rams.colostate.edu (M.R.); gabriele.landolt@colostate.edu (G.L.); dean.hendrickson@colostate.edu (D.H.)

**Keywords:** equine, celiotomy, postoperative complications, antimicrobials

## Abstract

**Simple Summary:**

Abdominal surgery is commonly performed in horses to treat gastrointestinal lesions or inflammatory conditions. Potential postoperative complications such as incisional infection have been shown to be reduced with appropriate perioperative antimicrobial administration. However, a recent survey of board-certified specialists on antimicrobial practices in horses undergoing celiotomy demonstrated that usage patterns were highly variable amongst practitioners and, at times, not concordant with the current literature. Furthermore, in light of recent evidence supporting an increase in antimicrobial resistance in equine veterinary practice, periodic reconsideration of antimicrobial regimens for commonly performed procedures is indicated. The goal of this retrospective study was to provide an updated review of the effect of antimicrobial protocols and other perioperative factors on postoperative complications in horses undergoing celiotomy at a referral hospital with the goal of advancing practitioners’ understanding of best practices and opportunities to reduce complications.

**Abstract:**

Recognition of antimicrobial resistance in equine practice has increased over the past decade. The objective of this study was to provide an updated retrospective review of antimicrobial regimens in one tertiary referral hospital and to evaluate the association with postoperative complications. A secondary objective was to evaluate other perioperative factors including surgical procedure, anesthetic and recovery parameters, and the effect of perioperative medications on complications and outcomes. A computerized search of medical records was performed to identify horses undergoing exploratory celiotomy from 1 January 2008 to 31 December 2021. A total of 742 celiotomies were performed (608 completed, 134 terminated intraoperatively). Factors recorded were evaluated using logistic regression for the presence of either incisional infection, postoperative ileus, or other complications postoperatively. Antimicrobial type or timing (pre-, intra-, or postoperative) were not associated with decreased risk of incisional infection or postoperative ileus; however, the duration of NSAID use was positively associated with incisional infection (OR 1.14 per day). Lidocaine and alpha-2-agonist administration postoperatively were also associated with increased incidence of postoperative ileus (OR 21.5 and 1.56, respectively). Poor recovery quality (OR 4.69), the addition of other antimicrobials besides penicillin/gentamicin postoperatively (OR 3.63), and an increased number of different NSAID classes used (OR 1.46 per additional) were associated with other complications. Implementation of enterotomy was associated with decreased risk of other complications (OR 0.64). These findings provide an updated summary of factors associated with postoperative complications in horses undergoing celiotomy.

## 1. Introduction

Exploratory celiotomy is commonly performed in equine practice to address obstructive or strangulating gastrointestinal lesions or inflammatory conditions such as peritonitis. Abdominal surgery has been demonstrated to be a risk factor for postoperative complications, including *Salmonella* infection, surgical site infection, thrombophlebitis, pneumonia, peritonitis, and adhesions, which in some instances have been shown to be reduced with appropriate antimicrobial protocols [1,2,3]. However, previous studies reviewing antimicrobial prophylaxis at a tertiary University referral hospital indicated that the majority of horses presenting for surgical colic received inaccurate antimicrobial prophylaxis in terms of timing of drug administration and dose received [4]. A more recent survey of board-certified equine specialists further reported that antimicrobial usage patterns were highly variable among practitioners and, at times, not concordant with the current literature [5]. Exploratory celiotomy is generally considered a ‘clean-contaminated’ procedure, with a small percentage of cases being classified as either ‘dirty’ (i.e. intra-abdominal contamination occurs) or ‘clean’ (i.e. performed on a non-emergent basis and without resection or enterotomy) [4,6]. Current recommendations for ‘clean-contaminated’ procedures include that broad-spectrum perioperative antimicrobial prophylaxis is indicated and recommended to be administered intravenously within 30 to 60 min prior to first incision, that aseptic technique should be followed intraoperatively, and that antimicrobials should be redosed intraoperatively if the procedure lasts longer than two half-lives [4,7]. Other factors related to case management within the surgeon’s control that have also been demonstrated to play a role in reducing surgical site infection include closure techniques, subcutaneous lavage, abdominal stenting, bandaging, and duration of antimicrobial administration postoperatively [8].

Recent evidence has called into question appropriate duration of antimicrobial administration following exploratory celiotomy in horses [6,9]. General guidelines in veterinary medicine support the concept that antimicrobials should be administered for the shortest effective duration to minimize development of resistant pathogens [10,11]. In humans, antimicrobial use beyond 24 h postoperatively has not been shown to reduce the risk of surgical site infection for clean-contaminated or dirty procedures [12]. The literature in equine practice further supports the concept that prolonged antimicrobial usage (single prophylactic dose or 72 h versus 120 h) did not yield additional benefits; furthermore, prolonged administration is known to contribute to development of complications such as colitis and *Salmonella* shedding [6,9,11]. However, whether these findings can be extrapolated to horses undergoing more extensive procedures (e.g., resection and anastomoses) without increased risk for complications such as peritonitis has not been reported to the authors’ knowledge, and requires further investigation. Another potential consideration in determining antimicrobial duration postoperatively is regarding prevention of intra-abdominal adhesions as one study in foals with experimental ischemia indicated that the combination of antimicrobials and anti-inflammatories administered for 72 h reduced adhesion risk, although shorter duration treatment protocols were not assessed [13]. Given recent concern for increased antimicrobial resistance in equine practice, and the expense of prolonged antimicrobial treatment, greater consideration of antimicrobial protocols tailored to specific surgical indications is warranted [14,15,16,17,18].

Therefore, the objective of this study was to provide an updated retrospective review of antimicrobial use in horses undergoing celiotomy procedures over the past ten years at one tertiary referral hospital and correlate the antimicrobial protocols with respect to the specific procedures (e.g., enterotomy, resection) with postoperative complications. As a secondary aim, we examined other intra- and perioperative factors, including procedure performed, medications administered, and anesthetic variables recorded, to determine if there were associations with postoperative complications. We hypothesized that prolonged antimicrobial administration past 72 h would not result in decreased postoperative complications, regardless of the type of procedure performed, but that other factors, including procedure performed and quality of anesthetic recovery, would be associated with complication risk.

## 2. Materials and Methods

### 2.1. Retrospective Review of Case Records

A computerized search of medical records was performed to identify horses undergoing exploratory celiotomy (completed and terminated prior to recovery) at Colorado State University within the last fourteen years (1 January 2008 to 31 December 2021). Medical records were retrospectively reviewed and data collected included patient and client name; surgeon; date; diagnosis; days hospitalized; survival to dismissal; whether enterotomy or bowel resection was performed; whether the horse was recovered from general anesthesia; anesthesia time; recovery time; recovery quality; whether incisional infection developed (yes/no); whether postoperative ileus was noted (yes/no); whether other complications were noted (yes/no) and, if so, what complications (e.g., jugular vein thrombophlebitis, recurrent colic or apparent abdominal pain, whether *Salmonellosis* shedding was noted during hospitalization); antimicrobial protocols administered; nonsteroidal anti-inflammatory administration (NSAID) (selection, dosing, duration); and whether other treatments were administered postoperatively (yes/no) and, if so, which treatments (e.g., prokinetics, opioids, gastroprotectants, alpha-2 agonists).

Categorization of cases that had complications (e.g., incisional infection) was based on the problem list stated in the medical record identified retrospectively. Briefly, complications were defined as previously described by Mair and Smith [19]. Surgical site infection was considered the presence of purulent discharge associated with swelling, heat, and pain around the skin incision [19]. Diagnosis of jugular vein thrombophlebitis was made by observation of clinical signs, including swelling over the affected vein +/− occlusion and findings on ultrasound [19]. Postoperative ileus was defined as a functional complication of surgery in horses with >2 L net reflux obtained through a nasogastric tube that did not have mechanical obstruction postoperatively [19]. Postoperative colic or evidence of abdominal pain was recorded in cases where the horse was observed to lie down for excessive periods, be inappetent, restless, flank-watch, stretch out as if to urinate, kick at the abdomen, sweat, paw, or roll [19]. Colitis or diarrhea was defined as persistent if greater than 24 h duration associated with pyrexia, with or without concurrent leucopenia and neutropenia [19]. Postoperative peritonitis was diagnosed based on combined clinical signs of depression, pyrexia, and endotoxemia with variable abdominal pain, in association with abnormal peritoneal fluid (total nucleated cell count >100 × 10^9^/L with cytological evidence of free or phagocytosed bacteria) [19]. In cases where complications were identified via retrospective review of records, each case was evaluated to ensure that these criteria were met.

As the effects of antimicrobial administration on the main outcomes was deemed the focus of the initial data collection, antimicrobial data was coded several different ways to capture any nuances that may have been present in the data. Initially antimicrobial data was transferred directly from the medical records: which antimicrobials were administered, whether antimicrobials were administered preoperatively (yes/no), intraoperatively (yes/no), duration of postoperative administration in days (as the vast majority were administered antimicrobials postoperatively), and timing of administration relative to the start of surgery if administered intraoperatively. As penicillin and gentamicin (pen/gent) were the most common antimicrobials administered, pre-, intra-, and postoperative antimicrobials were then classified as either pen/gent or other. Finally, data were coded for administration of any antimicrobial pre- or intraoperatively (yes/no) and pen/gent alone vs. any other combination at any time perioperatively (yes/no). These additional coded variables were then compared against the predetermined outcomes of incisional infection, postoperative ileus, and ‘other’ complications.

### 2.2. Data Analysis

Data were collated into a single .csv file for import into R (version 4.3.1 “Beagle Scouts”) for analysis [20]. Data were summarized using the *describe* function from the package Hmisc to evaluate for variables of interest [21]. Bias reduced penalized logistic regression was then performed using the function *logistf* from the logistf package [22]. Each of the 3 primary and secondary outcomes of interest (incisional infection, postoperative ileus, and other complications) was modeled versus all predictors recorded. Backwards selection was then performed automatically using the *stepAIC* function from the MASS package, as well as by hand using the *drop1* function in logistf [23]. If no predictors were significant using a *p*-value of 0.05 in the initial modeling or when all factors were removed and the null model was deemed the most informative, alternate sets of models were tested for significance, including anesthesia variables, antibiotic variables, etc. The specific predictors evaluated are noted in the figures/results where applicable or presented. Final predictors (if present) are presented using odds ratios and confidence intervals. All code used for analyses are available on GitHub at https://github.com/gregg-g/celio_retro_LP_GG_23.

## 3. Results

### 3.1. Case Load Summary

Over the fourteen-year time frame examined, a total of 742 exploratory celiotomies were performed, with 608 surgeries completed and 134 terminated intraoperatively. Of the 608 completed surgeries, 514 horses (84.5%) survived to dismissal. Days hospitalized ranged from 1 to 42 days with a mean of 6.0 days. Anesthetic time ranged from 0.5 to 6.5 h with a mean of 2.4 h. Recovery time from general anesthesia ranged from 0.25 to 1.7 h, with a mean of 1.3 h. Subjective grading of recovery quality was available in 593/608 cases and was considered excellent in 15 cases, good in 327 cases, fair in 180 cases, and poor in 71 cases. Enterotomy was performed in 324/608 cases (53.5%), and bowel resection was performed in 114/608 cases (18.8%).

### 3.2. Complications

Postoperative complications considered in the final analysis included incisional infection, postoperative ileus, or ‘other’ complications, which included recurrent or persistent colic, fever, endotoxemia, sepsis, peritonitis, hemoabdomen, laminitis, jugular vein thrombosis, cholangiohepatitis, leukopenia, coagulopathy, azotemia, anemia, anastomosis failure, colitis including *Salmonellosis* and *Clostridrium difficile* shedding, cardiac complications, musculoskeletal injury in recovery, and corneal ulceration. Incidence of incisional infection was recorded in 39/608 cases (6.4%). Incidence of postoperative ileus was recorded in 127/608 cases (20.9%). Other complications were recorded in 242/608 cases (39.8%).

### 3.3. Association of Perioperative Factors with Complications and Survival to Dismissal

The effect of antimicrobial regimen, administration of other medications, and perioperative factors were modeled for three outcomes: incidence of incisional infection, postoperative ileus, and other complications. Figure 1 shows final model factors associated with postoperative complications and their odds ratios. There were no significant effects on the measured outcomes when evaluating pre-, intra-, or postoperatively administered antimicrobials, with the exception of the administration of additional antimicrobials to penicillin and gentamicin postoperatively increasing the risk of additional complications. When NSAIDs were modeled by number of days administered postoperatively, there was a significant positive association with incisional infection, and administration of an increased number of different NSAIDs was associated with other recorded complications as well. Administration of lidocaine continuous rate infusion and alpha-2-agonists postoperatively were significantly associated with an increased incidence of postoperative ileus. A recovery from general anesthesia graded as poor versus excellent was predictive for other types of complications. Implementation of an enterotomy intraoperatively was associated with a decrease in other complications. Longer duration of hospitalization was weakly associated with a higher likelihood to survive to dismissal (*p* = 0.086), with an odds ratio (CI) of 1.058 (0.993–1.141) per day.

### 3.4. Antimicrobial regimens

Preoperative antimicrobial administration was recorded as administered in 205/608 cases (33.8%), with 10 different antimicrobial combinations noted, of which penicillin/gentamicin (pen/gent) was the most common in 186/205 cases (90.7%). Antimicrobials were administered intraoperatively in 171/608 cases (28.1%), with 17 different combinations being noted, with pen/gent again being the most common (74/171, 43.3%), followed by penicillin alone (20/171, 11.7%) and gentamicin alone (15/171, 8.8%). Duration of postoperative antimicrobial administration varied widely from no postoperative administration (5/608, 0.8%) up to 57 days duration (1/608, 0.2%), with a mean of 3.9 days. Additional drug classes of antimicrobials (besides those administered initially preoperatively, most commonly pen/gent) were given in 244/608 cases (40.1%). The odds ratios associated with various antimicrobial protocols and factors on the outcomes of interest are shown in Figure 2. While several of the outcomes have odds ratios that do not include ‘1’ (often indicating significance), the R^2^ for these models was between 0.002 and 0.017, indicating that these factors alone explain very little of the data.

### 3.5. Other Perioperative Medications

Duration of postoperative nonsteroidal anti-inflammatory (NSAID) administration ranged from 0 to 57 days with a mean of 4.3 days. The number of NSAIDs administered postoperatively ranged from 0 to 3 types, with one (424/605 or 70.1%) or two (168/605 or 27.8%) drug types being most common. Lidocaine continuous rate infusion was administered postoperatively in 501/606 cases (82.7%), for which that information was recorded. Alpha-2-agonists were administered postoperatively in 277/608 cases (45.6%). Opioids (i.e., butorphanol) were administered postoperatively in 221/608 (36.3%). Ketamine was administered postoperatively in 148/608 cases (24.3%).

## 4. Discussion

Selection and duration of antimicrobial prophylaxis remains a controversial aspect of equine surgical practice [24]. This study represents an updated reporting of complications following equine exploratory celiotomy with respect to comparing perioperative antimicrobial protocols and other aspects of case management. The main findings of this study did not support a difference in incidence of incisional infection, postoperative ileus, or other complications with different antimicrobial protocols used, pre- versus intraoperative dosing, or extended duration antimicrobial regimens beyond the perioperative period. However, the additional administration of other antimicrobial and NSAID drug classes was associated with an increased incidence of complications. Finally, several key outcomes noted in terms of perioperative case management that may be useful to practicing clinicians were identified, including the observation that integration of enterotomy procedures was associated with reduced postoperative complications (e.g., recurrent colic) and that poor recovery quality was predictive of developing other complications.

Current standards of care in equine veterinary surgical antimicrobial prophylaxis (SAP) were recently reviewed by Southwood et al. [25]. Resistant microbial infections have resulted in more than 1.2 million human deaths worldwide in 2019 [26]. With potentially increasingly limited options in the human medical field to treat active infection, the veterinary medical community has a responsibility for heightened awareness of methods to preserve antimicrobial function [27]. Furthermore, antimicrobial drug resistance is not only a concern in the human medical field, but reports in equine veterinary medicine exist describing multidrug resistant infections with *Staphylococcus*, *Escherichia coli*, and *Enterococci*, as well as methicillin-resistant *Staphylococcus aureus* (MRSA), vancomycin-resistant *Enterococci*, and rifampin- and macrolide-resistant *Rhodococcus equi* [28,29,30,31,32,33,34,35,36,37,38]. The impact of antimicrobial resistance on equine practice specifically, challenges to appropriate use, and the opportunity to improve clinical outcomes through responsible antimicrobial use, including in critical patients, has been emphasized in several pertinent recent review articles [25,39,40,41,42,43,44,45]. Factors contributing to appropriate antimicrobial use, such as indication for administration, selection, dosing, timing, route, duration, modification, and therapy have been reviewed recently by Hardefeldt et al. [46]. Emerging data generally support a shorter duration of SAP and encourage further development of collaborative antimicrobial stewardship programs within institutions [25]. Findings of this study are in concordance with this work, indicating that antimicrobial type or timing (pre-, intra-, or postoperative) were not associated with decreased risk of incisional infection or postoperative ileus, and that additional doses of antimicrobials did not affect complication rate. Multiple studies have demonstrated that antimicrobial drug administration and hospitalization increase antimicrobial drug resistance [33,47,48,49,50], and furthermore, that prolonged SAP is detrimental to the patient and increases costs associated with surgery [12]. These findings have led various groups (including the AVMA, AAHA, and others) to conclude that antimicrobial administration should be limited to cases in which the risk of infection is greater than 5%, which may encompass some cases of equine celiotomy such as those receiving resection and anastomoses or limited to treatment of infection and not used for infection prevention [7,51,52,53,54].

While clinical practice guidelines for SAP for human surgical patients have been well defined by the Surgical Infection Society, Infectious Disease Society of America, American Society of Health-System Pharmacists, and Society for Healthcare Epidemiology of America [47], there are fewer studies in equine surgery [25]. When compared to musculoskeletal procedures such as arthroscopy, exploratory celiotomy in horses tends to have a higher complication and infection rate [55]. A recent survey of antimicrobial use for equine celiotomy by Rockow et al. in 2023 showed little difference from a similar survey performed in 2002 by Traub-Dargatz et al., which indicated that most surgeons used penicillin and gentamicin for 24 h for standard exploratory laparotomy procedures, but implemented extended SAP for 3 to 5 days for horses undergoing enterotomy/resection or cases with large colon volvulus [5,56]. As mentioned, and previously supported by surgical prophylaxis studies in humans, antimicrobials administered beyond the operative period may be unnecessary in equine gastrointestinal surgery. These findings are supported by several additional equine studies supporting no difference in outcomes in 24 h SAP versus 3 or 5 days of extended prophylaxis or any postoperative antimicrobials following colic surgery [6,9,11]. The previous literature in this area, summarized recently by Southwood et al., and the findings of the current study support the concept that factors besides prolonged antimicrobial prophylaxis, such as intraoperative contamination, incision length, and postoperative colic are likely more important contributing factors to complications such as surgical site infection. Furthermore, periodic audits of antimicrobial use are indicated to determine whether clinical practices are congruent with recommended guidelines, as have been put forth here and by several groups recently [57,58,59]. However, it is recognized that care must be taken when using primarily retrospective data to drive how clinical medicine should be practiced, and until prospective case–control studies are performed, only inferences can be made based on the population sampled retrospectively. Ongoing prospective clinical trials in equine surgery will help to inform surgical practice moving forward in this era of increasing antimicrobial resistance.

The rate of incisional infection reported here (6.4%) compared favorably to previous reports of surgical site infection (SSI) prevalence following exploratory celiotomy (10–42%) [9,60,61,62,63]. This may be due in part to differences in definitions used in reporting SSI between previous publications and lack of follow-up available in some cases in this retrospective review. Antimicrobial protocols employed did not affect SSI rate in this case series, but it is further acknowledged that detecting statistical differences would be difficult due to the low infection rate overall. Of the perioperative variables assessed, only duration of NSAID usage postoperatively was positively associated with an increased incidence of SSI, with a multiplicative odds ratio of 1.14 per day. This was attributed to continued administration due to observed inflammation associated with the incision site, and was not presumed to be causative for SSI in this population.

Postoperative ileus was recorded in 20.9% of cases in this retrospective study, which is similar to recent reviews reporting 18.4 to 33% incidence of ileus following celiotomy [64,65,66,67,68,69]. Variables previously associated with development of ileus have been the presence of nasogastric reflux (>8 L) preoperatively, elevated heart rate, elevated packed cell volume and hyperglycemia at admission examination, increased age, small intestinal lesions, resection and anastomosis performed, and increased duration of anesthesia [65,67,69,70], variables which were not comprehensively assessed in this study. Aspects of case management that were associated with ileus in this case population were administration of lidocaine continuous rate infusion and alpha-2-agonists postoperatively, which, as with NSAID administration in the case of SSI above, were presumed to be continued in response to observed POR and not causative. Treatment and prevention of ileus involves supportive care, including prokinetic drugs such as lidocaine, metoclopramide, cisapride and erythromycin [65,66,67,68,69]. Although lidocaine is administered based on clinician preference in this tertiary referral hospital, it was prescribed in the majority of cases undergoing celiotomy during this time frame (82.7%).

Throughout the literature, there are conflicting reports regarding the efficacy of lidocaine as a prokinetic, with no effect on ileus or survival reported in the UK population [69] and with evidence to the contrary demonstrating lidocaine improved postoperative ileus, survival rate, time to first manure passage, and duration of hospitalization in other studies [65,71]. However, the majority of both the European Colleges of Veterinary Surgeons/Equine Internal Medicine clinicians (79%) and American Colleges of Veterinary Surgeons/Internal Medicine Clinicians/Emergency Critical Care diplomates (68%) have continued to report using lidocaine to treat horses with postoperative ileus [72,73]. A recent meta-analysis evaluating the available body of literature regarding efficacy of lidocaine to reduce ileus concluded that lidocaine was associated with an increased incidence of diagnosis of ileus (as in our study), and that horses treated with lidocaine were more likely to survive to dismissal, but that lidocaine administration was not specifically associated with reduced postoperative ileus [39]. These findings suggest further investigation of lidocaine as a treatment specifically for ileus is necessary [74], but these data did not show an association with successful discharge from the hospital with lidocaine use. The use of alpha-2 agonists postoperatively, however, increased the risk of ileus whether lidocaine was being administered or not, as the odds ratios are calculated based on holding all other factors constant. This is consistent with the significant reduction in gastrointestinal motility seen with the use of all studied alpha-2 agonists [75].

In this study, complications other than SSI or postoperative ileus were evaluated together and recorded in 39.8% of cases. Interestingly, inclusion of enterotomy at time of surgery was found to significantly reduce the risk of other complications. As postoperative colic was the most common ‘other’ complication recorded, this finding is postulated to be primarily associated with reduced risk of recurrent colic following enterotomy due to resolution of large colon impaction following pelvic flexure enterotomy. It is further the authors’ clinical experience that patients may be refed more rapidly postoperatively and have reduced need for repeat celiotomy following evacuation of bowel contents with enterotomy in either primarily large colon cases or small intestinal strangulating lesions with secondary large colon impaction. Additionally, anesthetic recoveries that were graded as poor versus excellent were predictive for a higher risk of postoperative complications. Equine anesthesia is generally associated with a greater risk of morbidity and mortality compared to humans and companion animals (1% versus 0.001–0.1%) [76,77,78] and horses presenting for colic or other emergency surgery, for procedures occurring outside regular working hours or for longer anesthetic periods, which is often the case for emergency celiotomies, have been previously reported to have a higher risk of mortality associated with general anesthesia [77,78,79,80,81,82]. Due to the retrospective nature of this study, it is not possible to distinguish whether recoveries assessed as poor resulted simply from the severity of disease status in which case a higher rate of complications could be anticipated or whether recovery quality itself affected complication rate (e.g., musculoskeletal injury, corneal laceration), or a combination of both. However, these findings provide potentially actionable information to clinicians in terms of case management intraoperatively and communication with clients regarding the risk of complications postoperatively and should provide additional impetus to improve anesthetic recoveries whenever possible in all situations.

Limitations of this study include the retrospective nature of study design, which limited retrieval of full data sets for all surgeries performed. Active questioning of all clients regarding complications that occurred following dismissal from the hospital was not performed. The complications evaluated individually statistically were those seen most frequently in this case population while others were grouped together to facilitate statistical analyses. The low rate of incisional site infection overall may have prohibited evaluation of individual antimicrobial medication regimens as factors associated with resultant infection. The consistent use of antimicrobials in all cases, whether started pre-, intra-, or postoperatively, also precludes evaluating the necessity of antimicrobial administration in this population. As owners were not directly contacted, it is possible that additional follow-up would have allowed discovery and recording of more complete information regarding postoperative outcomes and complications, and that the complication rate reported here may be underestimated. Furthermore, the analysis of ‘other’ complications as a single group may have prohibited ability to detect differences specific to each type of complication. Additional prospective studies examining the effect of drug administration versus time on postoperative complications are indicated.

## 5. Conclusions

In summary, antimicrobial protocol did not affect postoperative complication rate in this study population of horses undergoing exploratory celiotomy, with no additional benefit observed to antimicrobial administration beyond the perioperative period. The addition of antimicrobial classes besides penicillin and gentamicin and multiple NSAIDs was associated with an increased risk of complications. Longer NSAID duration was associated with an increased risk of incisional infection. Lidocaine and alpha-2-agonist administration were associated with postoperative ileus. Periodic audits of clinical practices associated with frequently performed procedures such as exploratory celiotomy are indicated to reevaluate current practices in light of the recent literature and to potentially improve outcomes. These data may be useful to surgeons in evaluating their own postoperative complication rates and in assessing factors associated with increased risk of complications following exploratory celiotomy.

## Figures and Tables

**Figure 1 animals-13-03573-f001:**
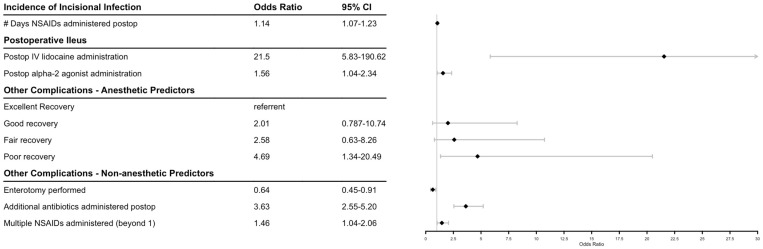
Forest plot of factors associated with equine celiotomies. Diamonds indicate odds ratio.

**Figure 2 animals-13-03573-f002:**
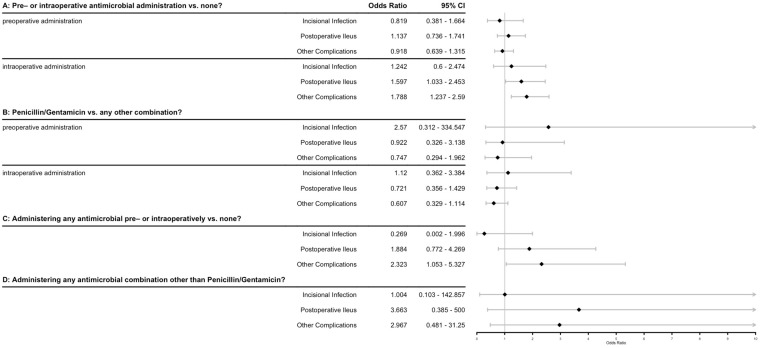
Forest plot of antimicrobial factors associated with various outcomes after equine celiotomies. Odds ratios and confidence intervals are shown for individual logistic regression models using the 3 primary outcomes of interest (incisional infection, postoperative ileus, other complications) as the dependent variables. The 4 independent variables were defined as questions: A: Does the administration of any antibiotic either pre- or intraoperatively (modeled separately) have an effect on the probability of the outcomes of interest? B: Does the administration of penicillin/gentamicin vs. any other antimicrobial combination pre- or intraoperatively (modeled separately) have an effect on the probability of the outcomes of interest? C: Does the administration of any antimicrobial at all in the pre- or intraoperative period affect the probability of the outcomes of interest? D: Does the administration of any antimicrobial combination other than penicillin/gentamicin at any time in the perioperative period (pre-, intra- or post-) affect the probability of the outcomes of interest? Diamonds indicate odds ratio.

## Data Availability

Data available in a publicly accessible repository as noted in the manuscript.

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
