# Peer review of "The Effects of Antimicrobial Protocols and Other Perioperative Factors on Postoperative Complications in Horses Undergoing Celiotomy: A Retrospective Analysis, 2008–2021"

_animals, 2023, doi:10.3390/ani13223573_

Round 1
Reviewer 1 Report
Comments and Suggestions for Authors
General comment
Well written manuscript, clear, concise. However, it is so concise that some data of interest seem to be missing and the overall impression is more of a short communication than a standard article.
Major comments
The title starts with “Effect of Antimicrobial Protocol … ” but it is not clear how all parameters mentioned in the methods were analyzed. In reference to “Antibiotics protocols (which antibiotics, timing of administration relative to surgery, whether antibiotics were administered intraoperatively (yes/no), duration of antibiotics postoperatively)”, please indicate how the type of antibiotics was analyzed (penicillin-gentamicin versus “other” or penicillin-gentamicin versus “other” versus "unknown" ?). Please also clarify how the duration was analyzed (ex: stratification or odds ratio for additional day of treatment, ...). Since it seems to be the core of the study (starts the title, intro, discussion), it would be more interesting to provide the results associated with the antibiotics protocols in full even if they are not statistically significant.
Antimicrobial regimens/protocols: The proportion of missing data for the type of preoperative antimicrobials administered is staggering (66%) and limits a major component of the comparison between protocols. The reviewer understands the limitations of retrospective studies but this seems to be very basic information that is required in a medical file and can / should be found (anesthesia file, billing file, pharmacy, …). This raise questions on the validity of the other data recorded, especially the ones that are “softer” such as the presence of an infection. Please make sure that medical records were thoroughly reviewed and address/explain the missing antimicrobials data and how it can impact the conclusions.
Associations and Figure 1: The reviewer is not a statistician but it is my understanding that in Forest plots, factors for which the 95% CI crosses the line of the null effect do not reflect a statistically significant result. Please add p values and review the result section and discussion accordingly.
Please provide a rationale for the sample size selected and a retrospective power analysis for the main outcome taking in account the rate of infections and the missing antibiotics data. Please adjust the conclusion accordingly.
It is unclear why the type of lesion was not incorporated in the analysis (ex: small intestine vs large). It is especially of interest in the light of the association with NSAIDs and lidocaine.
Other comments
Simple summary:
Please remove “other” before inflammatory conditions. It suggests that colics are inflammatory conditions.
Introduction
“Postoperative complications (…) have been shown to be reduced with appropriate antimicrobial protocols [1-4].” Data in reference 1 do not support this sentence. Reference 4: since all horses that underwent surgery received antimicrobials, it is unlikely that this reference has data supporting this sentence. Please review carefully each of these references and remove this sentence if it cannot be backed up by published data. In general, please make sure that the references cited have studied the point made and are not just mentioning it in the introduction.
Methods
Please provide more information on the definitions used for incisional infection, reflux, ileus and thrombophlebitis. Was it only based on the problem list or files were reviewed to make sure there was somewhat of a common definition (ex: reflux X L once or more than X liters over X period of time).
Please indicate the level of significance used.
Results
Antimicrobial regimens: “Additional drug classes (…) were given in 244/608”. Since the type of preoperative antimicrobials were not known for 403 cases, it seems that the denominator here should be 205, not 608.
“Longer duration …” results that are not statistically significant don’t have “higher likelihood”. Please rephrase. Also, please indicate how “longer” is defined for this analysis.
Author Response
Effect of antimicrobial protocol and other perioperative factors on postoperative complications in horses undergoing celiotomy: a retrospective analysis 2008-2021
Editor comments
- Please kindly provide the first author's brief curriculum vitae (CV) file
(previous email and attachment).
- This has been attached.
Please kindly increase the number of words in the main text to over 4000.
- The word count has been increased to approximately 4400 and each comment addressed below.
Please check that all references are relevant to the contents of the manuscript.
- References have been verified and additional references added as indicated.
Any revisions to the manuscript should be highlighted, such that any changes can be easily reviewed by editors and reviewers.
- Revisions have been tracked throughout the revised version.
Please provide a cover letter to explain, point by point, the details of the revisions to the manuscript and your responses to the referees’ comments.
- Comments have been addressed individually below.
If you found it impossible to address certain comments in the review reports, please include an explanation in your appeal.
- All comments have been addressed individually below.
The revised version will be sent to the editors and reviewers.
- We thank the editors and reviewers for their constructive comments and look forward to receiving further feedback on the manuscript.
Reviewer 1
Well written manuscript, clear, concise. However, it is so concise that some data of interest seem to be missing and the overall impression is more of a short communication than a standard article.
- Thank you for this feedback. Additional data have been added and the discussion expanded upon.
Major comments
The title starts with “Effect of Antimicrobial Protocol …” but it is not clear how all parameters mentioned in the methods were analyzed. In reference to “Antibiotics protocols (which antibiotics, timing of administration relative to surgery, whether antibiotics were administered intraoperatively (yes/no), duration of antibiotics postoperatively)”, please indicate how the type of antibiotics was analyzed (penicillin-gentamicin versus “other” or penicillin-gentamicin versus “other” versus "unknown"?).
- You are correct that the analysis was summarized significantly. We have added a paragraph (lines 113 – 126) and second figure to further expand on the evaluation of the antibiotic protocols.
Please also clarify how the duration was analyzed (ex: stratification or odds ratio for additional day of treatment, ...).
- Duration postoperatively was by day as now noted in the materials and methods (now line 118).
Since it seems to be the core of the study (starts the title, intro, discussion), it would be more interesting to provide the results associated with the antibiotics protocols in full even if they are not statistically significant.
- The results of the various antibiotic protocols and analyses are presented in the new Figure 2. These were not initially included as the modelling did not demonstrate these as significant predictors. The new Figure, while demonstrating some ‘significant’ predictors, is a bit misleading as the antibiotic protocols explain very little of the overall variability in the data (less than 2%). This was added to the results (lines 176 – 179). Antibiotic protocols and antibiotic resistance patterns noted in equine practice have been further expanded upon in the discussion section.
Antimicrobial regimens/protocols: The proportion of missing data for the type of preoperative antimicrobials administered is staggering (66%) and limits a major component of the comparison between protocols. The reviewer understands the limitations of retrospective studies, but this seems to be very basic information that is required in a medical file and can / should be found (anesthesia file, billing file, pharmacy, …). This raise questions on the validity of the other data recorded, especially the ones that are “softer” such as the presence of an infection. Please make sure that medical records were thoroughly reviewed and address/explain the missing antimicrobials data and how it can impact the conclusions.
- Our apologies, as this was a misrepresentation of the data. While it may appear that 66% of the data is missing, this in in fact the percentage of horses that did not receive pre-operative antibiotics (some were given following induction/intraoperatively due to the severity of pain experienced by the horse. This has been clarified in the manuscript (lines 166 – 171)
Associations and Figure 1: The reviewer is not a statistician but it is my understanding that in Forest plots, factors for which the 95% CI crosses the line of the null effect do not reflect a statistically significant result. Please add p values and review the result section and discussion accordingly.
- Thank you for noticing that – the reviewer is correct that, with very few exceptions, odds ratios that include ‘1’ are not strictly significant. However, we do not generally use p-values in regression analysis for presentation of the final models, as these can bring a very ‘black and white’ evaluation of the data. The choice of factors to leave in the final model is based on p-values, AIC, BIC, and physiologic plausibility – to assume that the p-value is the constraint leaves a lot of information on the table. In addition, the reliance on p-values for regression models does not allow for individual evaluation of the odds ratios. If the odds ratio for adverse outcomes after a treatment is (for example) 0.99 – 27.4, a risk tolerant individual might assume that they could continue that treatment, as the risk of poor outcomes might be the same, while a risk-adverse individual would likely assume that the risk of adverse events is probably higher than 1 – but both seeing a p-value of 0.06…
If the reviewer feel strongly that the p-values should be included (even though they were not discussed in the text), we can add those to the table. The discussion did not reference the p-values, and specifically noted the ‘strength’ of association of the outcomes and measured variables for reasons noted above. We have removed the word significant (line 195) and replaced it with ‘final model factors’ so as not to confuse the reader.
Please provide a rationale for the sample size selected and a retrospective power analysis for the main outcome taking in account the rate of infections and the missing antibiotics data. Please adjust the conclusion accordingly.
- We cannot provide you with a power analysis. The sample size was not calculated – a date range selection was made based on what could be reasonable evaluated in the time available while taking into account changes in clinical practice. While a power analysis could be done with single factor analysis, the number of factors as well as the variability in predictors between outcomes precludes any power analysis in this multifactorial setting.
It is unclear why the type of lesion was not incorporated in the analysis (ex: small intestine vs large). It is especially of interest in the light of the association with NSAIDs and lidocaine.
- We agree that the type of lesion could be of interest, but the number of surgery reports that reported both, in conjunction with the number of surgeries that involved both small and large intestine (i.e., primary small intestinal resection with secondary large colon impaction necessitating pelvic flexure enterotomy), precluded categorization in this manner. The authors felt this could not be resolved retrospectively by a record review and thus was not included.
Other comments
Simple summary:
Please remove “other” before inflammatory conditions. It suggests that colics are inflammatory conditions.
- Agreed, the word was removed.
Introduction
“Postoperative complications (…) have been shown to be reduced with appropriate antimicrobial protocols [1-4].” Data in reference 1 do not support this sentence. Reference 4: since all horses that underwent surgery received antimicrobials, it is unlikely that this reference has data supporting this sentence. Please review carefully each of these references and remove this sentence if it cannot be backed up by published data. In general, please make sure that the references cited have studied the point made and are not just mentioning it in the introduction.
- Thank you for this suggestion – these references and their order in the manuscript have been revised.
Methods
Please provide more information on the definitions used for incisional infection, reflux, ileus and thrombophlebitis. Was it only based on the problem list or files were reviewed to make sure there was somewhat of a common definition (ex: reflux X L once or more than X liters over X period of time).
- Thank you for requesting this information to be added. Definitions of the above have been provided and cited to previous literature.
Please indicate the level of significance used.
- This depended on the analysis – most of the analyses used AIC, not strict significance (noted in the analysis section). We have added some information in the analysis section on how or why factors were included or removed from the models (lines 137-138). When significance was necessary or perhaps not what would be considered normal (such as for days hospitalized), the p-value for the result was noted in the text.
Results
Antimicrobial regimens: “Additional drug classes (…) were given in 244/608”. Since the type of preoperative antimicrobials were not known for 403 cases, it seems that the denominator here should be 205, not 608.
- Thank you for noticing this, you are correct that this number was transcribed wrong. The correct information is presented in the text in lines 169 – 173.
- Antimicrobial regimens - Preoperative antimicrobial administration was recorded as administered in 205/608 cases (33.8%), with 10 different antibiotic combinations noted of which penicillin/gentamicin was the most common in 186/205 cases (90.7%). Antimicrobials were administered intraoperatively in 171/608 cases (28.1%), with 17 different combinations being noted, with pen/gent again being the most common (74/171, 43.3%) followed by penicillin alone (20/171, 11.7%) and gentamicin alone (15/171, 8.8%).
“Longer duration …” results that are not statistically significant don’t have “higher likelihood”. Please rephrase. Also, please indicate how “longer” is defined for this analysis.
- We note that this was ‘weakly associated’ – the use of strict p-values in this sort of multivariable logistic regression can be problematic. As the p-value associated with this predictor is noted in the results, we would argue that the reader should be left with the evaluation of whether their specific risk tolerance/adverse profile makes this specific. The addition of ‘per day’ was added as per request.
Reviewer 2 Report
Comments and Suggestions for Authors
Based on the title of the paper most of the discussion has no merit on the use of antimicrobials. The authors explain about POR and the causation and whether lidocaine is a beneficial treatment option and how it can be controversial. While this is very true and its use is widely debated this is well beyond the scope of the title of this paper Effect of Antimicrobial Protocol.... Unfortunately the authors actually spend little time talking about any of the antimicrobial regimes at all. They mention that beta lactam amino glycoside was the most common but they failed to state what the second most common was but stated that there was one. Also unfortunately with this being a retrospective only 205/605 cases had recorded antimicrobial types given which makes drawing any usable data somewhat challenging. There is good merit to this paper but the discussion needs significant work to shift the focus of the paper back onto the title which involves antimicrobial administration.
Comments on the Quality of English LanguageThe English and gammer were appropriate.
Author Response
Reviewer 2
Based on the title of the paper most of the discussion has no merit on the use of antimicrobials. The authors explain about POR and the causation and whether lidocaine is a beneficial treatment option and how it can be controversial. While this is very true and its use is widely debated this is well beyond the scope of the title of this paper Effect of Antimicrobial Protocol.... Unfortunately the authors actually spend little time talking about any of the antimicrobial regimes at all. They mention that beta lactam amino glycoside was the most common but they failed to state what the second most common was but stated that there was one. Also unfortunately with this being a retrospective only 205/605 cases had recorded antimicrobial types given which makes drawing any usable data somewhat challenging. There is good merit to this paper but the discussion needs significant work to shift the focus of the paper back onto the title which involves antimicrobial administration.
- Thank you for this feedback and the reviewer’s time spent evaluating the manuscript. The authors have significantly expanded the methods, results and discussion sections surrounding and emphasizing the antimicrobial protocols used and their relation to postoperative complications to address the reviewer’s comments. References have been added and updated. The authors would maintain that the other information regarding additional perioperative factors is still relevant to the practicing clinician and reflected in the title and was therefore maintained in the discussion. The information regarding number of cases that received preoperative or intraoperative antimicrobials has been clarified in the results section. The discussion has been revised to emphasize and discuss further the antimicrobial data presented.
Round 2
Reviewer 1 Report
Comments and Suggestions for Authors
Thank you for your reviewed version. The authors have addressed the reviewer's concerns to satisfaction.
Author Response
Thank you to the reviewer for their time and constructive comments towards improving the manuscript
Reviewer 2 Report
Comments and Suggestions for Authors
The reviewer would like to thank the authors for resubmission of this manuscript.
Unfortunately I now have even more concerns about this manuscript because it seems that while effort was made to try to improve the paper it was done in a very rushed manner. Example lines 115 to 133 you choose to cite every complication from the same review manuscript while this is some what appropriate there are many manuscripts in publication that define POI, incisional infection etc. I would strongly suggest that you go into the literature and use those original sources rather than just one review article Also, can you now define why you chose to evaluate POI, pain, diarrhea, incisional complication, thrombophlebitis etc and not a significant list of other problems that any post operative horse could have, they appear to be just randomly selected.
Line 139 why did you choose to only evaluate time of post operative antimicrobial administration vs post operative antimicrobial administration vs length of time of post operative antimicrobial administration. You need to assess if there was a drug administration effect vs a time effect not just a time effect
Line 146 you talk about post op reflux do you mean POI please clarify and justify.
Line 146 please define how you came up with the predetermined outcomes of incisional infection, post op reflux and other. Why did you choose these vs all of the others that you listed in the paragraph above. There is no consistency in this part of the manuscript.
Line 188 Please list what drug combination was the most common in the post operative period and for what length of time not just length of time antimicrobials were prescribed. This is misleading because multiple different types and classes of antimicrobial agents may have been administered during this time.
Fig 2
Line 229-238 Please define how you formulated your logistic regression in M and M section not in the figure legend.
Line 273 Please indicate if you are interested in POI or post op reflux and define which you are interested in and why. They are different but you refer to both in this paper.
Line 278 Please clarify what do you mean by "various groups"
Line 279 A 5% risk is not very high this could be argued that every celiotomy that has an R/A or an enterotomy has a 5% risk of infection. This also could be said for any horse that has an intravenous catheter placed that there is a 5% risk of phlebitis that may develop into a septic thrombophlebitis
Line 291 -302 While there may be some limited evidence from these papers we need to be very care using mostly retrospective data as a driving force to speak for how clinical medicine should be practiced. Until prospective case control studies are published only inferences can be gathered based on the population that was sampled in the retrospective studies and cannot be applied in any manor in a prospective way as dogma for how treatments should be changed or practiced. Proper prospective studies need to be accomplished and there are several ongoing currently. While there is use in looking retrospectively the only information that can be taken away is what was seen in that group of animals at that point in time with the treatments that were applied to that group. The outcomes cannot and should not be prospectively assumed to be the expected outcomes in a new population of animals unless tested. Please remove and or revise this paragraph.
Comments on the Quality of English Language
No major issues.
Author Response
Reviewer 2
The reviewer would like to thank the authors for resubmission of this manuscript. Unfortunately I now have even more concerns about this manuscript because it seems that while effort was made to try to improve the paper it was done in a very rushed manner. Example lines 115 to 133 you choose to cite every complication from the same review manuscript while this is some what appropriate there are many manuscripts in publication that define POI, incisional infection etc. I would strongly suggest that you go into the literature and use those original sources rather than just one review article. Also, can you now define why you chose to evaluate POI, pain, diarrhea, incisional complication, thrombophlebitis etc and not a significant list of other problems that any post operative horse could have, they appear to be just randomly selected.
- Thank you to the reviewer for their time and consideration towards improving the manuscript. The authors have made every attempt to address each comment fully. The review article cited is the original source for these definitions (i.e., the authors have confirmed going back to the article that it does not provide additional references for these definitions). Furthermore, the review article is heavily cited as a reference text and the authors therefore would consider it appropriate for reference here. The reviewer raises a good point regarding selection of complications evaluated which was performed as such as they are those seen most commonly in this case population while the others were grouped together as ‘other’ to facilitate statistical analyses for those seen infrequently. This has been further discussed in the limitations section of the discussion section.
Line 139 why did you choose to only evaluate time of post operative antimicrobial administration vs post operative antimicrobial administration vs length of time of post operative antimicrobial administration. You need to assess if there was a drug administration effect vs a time effect not just a time effect.
- Thank you for noting this important point. Unfortunately, we were unable to evaluate the effect of postoperative antimicrobial administration (whether given or not), as only 5/607 were not administered antibiotics in the postoperative period (noted in lines 147-148 and again in lines 203 - 205). Recording the duration of postoperative antibiotic administration as number of days (0, 1, 2, etc.) more accurately captures this as a linear predictor with zero representing those not administered postoperative antibiotics. A prospective evaluation of the use of antibiotics postoperatively would need to be conducted to properly answer the reviewer’s question which is why we did not attempt to answer the question of whether antibiotics should be used in the postoperative period – we can only discuss the data we are able to analyze. This has been added to the discussion section.
Line 146 you talk about post op reflux do you mean POI please clarify.
- Thank you for requesting this clarification. Terminology has been altered to postoperative ileus throughout the manuscript and figures.
Line 146 please define how you came up with the predetermined outcomes of incisional infection, post op reflux and other. Why did you choose these vs all of the others that you listed in the paragraph above. There is no consistency in this part of the manuscript.
- These were chosen as primary outcomes as they were seen most commonly as complications in this case population while all others were grouped together in the ‘other’ category to facilitate statistical analyses due to their low frequency. This has been clarified in the text.
Line 188 Please list what drug combination was the most common in the post operative period and for what length of time not just length of time antimicrobials were prescribed. This is misleading because multiple different types and classes of antimicrobial agents may have been administered during this time.
- We have added a note that penicillin/gentamicin was the most commonly prescribed drug combination postoperatively (line 207). The reviewer is correct that there were a vast number of different drug combinations administered postoperatively, with just over 40% of horses receiving ‘other’ antimicrobials (lines 206-207).Many of these horses also received one combination for 2 days, then a different drug for a few days, then perhaps a third (or even 4th and 5th) combination for some number of days. In order to extract any useful information, the authors chose to classify this as number of days total, as well as defining pen/gent alone vs other various other combinations.
- If we were to try and pull other antimicrobial combinations as well as durations, when they were started and stopped, with what other antimicrobials, etc. there would be more factors than a model would be able to handle – thus the simplification into a smaller number of factors that could be modelled. We are not trying to be misleading, but rather to extract some useful information from what is a very complicated data set.
Fig 2
Line 229-238 Please define how you formulated your logistic regression in M and M section not in the figure legend.
- Thank you for noting this. Lines 171 – 174 of the M&M section note the reasons for evaluating this subset of variables. The full text of the “questions” evaluated is left in the figure legend as putting this into the figure itself left the data mostly unreadable – and we were more interested in the data instead of the text. A line was added to the M&Ms to note this – “The specific predictors evaluated are noted in the figures/results where applicable or presented.” (lines 174 – 175).
Line 273 Please indicate if you are interested in POI or post op reflux and define which you are interested in and why. They are different but you refer to both in this paper.
- Terminology throughout the manuscript has been clarified to postoperative ileus rather than reflux.
Line 278 Please clarify what do you mean by "various groups"
- This has been noted in the text (including the AVMA, AAHA and others) as well as 2 more references added.
Line 279 A 5% risk is not very high this could be argued that every celiotomy that has an R/A or an enterotomy has a 5% risk of infection. This also could be said for any horse that has an intravenous catheter placed that there is a 5% risk of phlebitis that may develop into a septic thrombophlebitis
- This sentence has been tempered to take into account the reviewer’s comment. References are provided which support that other groups have suggested antimicrobial administration should be limited to cases in which the risk of infection is greater than 5%.
Line 291 -302 While there may be some limited evidence from these papers we need to be very care using mostly retrospective data as a driving force to speak for how clinical medicine should be practiced. Until prospective case control studies are published only inferences can be gathered based on the population that was sampled in the retrospective studies and cannot be applied in any manor in a prospective way as dogma for how treatments should be changed or practiced. Proper prospective studies need to be accomplished and there are several ongoing currently. While there is use in looking retrospectively the only information that can be taken away is what was seen in that group of animals at that point in time with the treatments that were applied to that group. The outcomes cannot and should not be prospectively assumed to be the expected outcomes in a new population of animals unless tested. Please remove and or revise this paragraph.
- Thank you for these thoughtful comments. This paragraph has been modified to take into account the reviewer’s comments.